# Global Analysis of the WOX Transcription Factor Family in *Akebia trifoliata*

**Shengpeng Chen, Huai Yang, Yongle Zhang, Chen Chen, Tianheng Ren** [ID]**, Feiquan Tan and Peigao Luo *** [ID]

Key Laboratory of Plant Genetics and Breeding at Sichuan Agricultural University of Sichuan Province, Chengdu 611130, China; chenshengpeng618@163.com (S.C.); yanghuai202103@163.com (H.Y.); zhangyongle0301@163.com (Y.Z.); icbrcc2018@163.com (C.C.); renth@sicau.edu.cn (T.R.); tanfq121@aliyun.com (F.T.)

* Correspondence: lpglab@sicau.edu.cn

**Abstract:** *Akebia trifoliata* is an economically important, self-incompatible fruit tree in the Lardizabalaceae family. Asexual propagation is the main strategy used to maintain excellent agronomic traits. However, the generation of adventitious roots during asexual propagation is very difficult. To study the important role of the WUSCHEL-related homeobox (WOX) transcription factor in adventitious root growth and development, we characterized this transcription factor family in the whole genome of *A. trifoliata*. A total of 10 *AktWOXs* were identified, with the following characteristics: length (657~11,328 bp), exon number (2~5), isoelectric point (5.65~9.03), amino acid number (176~361 AA) and molecular weight (20.500~40.173 kDa), and their corresponding expression sequence could also be detectable in the public transcriptomic data for *A. trifoliata* fruit. A total of 10 *AktWOXs* were classified into modern (6), intermediate (2) and ancient clades (2) and all *AktWOXs* had undergone strong purifying selection during evolution. The expression profile of *AktWOXs* during *A. trifoliata* adventitious root formation indicated that *AktWOXs* play an important role in the regulation of adventitious root development. Overall, this is the first study to identify and characterize the *WOX* family in *A. trifoliata* and will be helpful for further research on *A. trifoliata* adventitious root formation.

**Keywords:** *Akebia trifoliata*; WUSCHEL-related homeobox; transcription factor; adventitious roots

## 1. Introduction

*Akebia trifoliata* (Thunb.) Koidz. (2n = 2x = 32) belongs to the flowering plant family Lardizabalaceae [1]. As the third generation of emerging fruit, the flesh of *A. trifoliata*, which is deeply loved by people, not only has a delicate texture and sweet taste but also contains many free essential amino acids [2]. Therefore, the artificial cultivation of *A. trifoliata* has been rapidly increasing in Southwest China and the middle and lower reaches of the Yangtze River in recent years. However, *A. trifoliata* breeding techniques are not yet able to meet the needs of farmers because *A. trifoliata* is a cross-pollinated plant and good maternal traits can only be maintained through asexual reproduction [3].

To date, research on asexual breeding methods for *A. trifoliata* has mainly focused on tissue culture breeding and cutting breeding. In the exploration of tissue cultures, Wu et al. (2015) established and optimized an efficient callus culture system using leaves as explants and established a method for the rapid propagation of stems with leaf buds, with induction and rooting rates of more than 80% in the optimized medium [4]. This method can shorten the seedling cycle of seedlings. However, since the formation of endogenous toxins and adventitious roots in *A. trifoliata* is hard to achieve, it is still difficult to establish a complete tissue culture system for *A. trifoliata* [5]. The method of culturing cuttings of *A. trifoliata* has also attracted the attention of researchers. Studies have shown that fine river sand + nutrient soil is the preferred medium for cuttings of *A. trifoliata* [6]. In addition, plant growth regulators are also used to promote the growth of *A. trifoliata* roots. There is a report showing that ABT2 rooting-powder treatment can effectively promote the rooting of

*A. trifoliata* cuttings [7]. In China, the *A. trifoliata* cutting system has slowly begun to mature. The unstable roots produced by cuttings and tissue culture are liable to fall off and do not easily survive field transplantation. The root system plays a crucial role in the growth and development of the whole plant. Therefore, the study of the formation and development process of adventitious roots helps us to obtain high-quality saplings *A. trifoliata* saplings.

The formation of adventitious roots is one of the key steps of plant asexual propagation [8], and the WUSCHEL-related homeobox (WOX) transcription factors widely present in plant genomes have been shown to be involved in the regulation of adventitious root formation [9]. For example, in *Arabidopsis thaliana*, *WOX11* and *WOX12* respond to auxin induction and then activate the expression of *WOX5* and *WOX7* to change the cell fate from root invasive cells to root primordium cells and achieve adventitious root regeneration [9,10]; in *Oryza sativa*, *OsWOX3A* leads to an increase in plant lateral root number, indicating that *OsWOX3A* may be involved in the regulation of GA-IAA crosstalk in rice root development [11]; in the gymnosperms *Picea-Abies* and *Populus nigra*, *PsWOX3* is expressed in a few cells on the peripheral surface of the shoot apical meristem, and *PaWOX3* is highly expressed in the root tip [12,13]. The overexpression of *MdWOX11* promotes adventive root primordium formation in apples, while the interference of *MdWOX11* inhibits adventive root primordium production [14]. Therefore, some members of the WOX transcription factor family play important roles in the growth and development of adventitious roots.

At present, genome-wide identification of the WOX transcription factor family has been completed in many plants. The WOX family is a group of plant-specific transcription factors and belongs to the homeobox (HB) transcription factor family. The typical homeodomain (HD) of the HB superfamily has 60–66 amino acid residues that fold into a "helix-loop-helix-turn-helix" spatial structure, where a combination of the second and third helices forms a "helix-turn-helix" that can bind to specific DNA sequences [15]. *WUSCHEL (WUS)* is the most primitive gene in the WOX transcription factor family. In 2004, Haecker et al. identified 14 other members with similar structures by using homologous methods to search for *A. thaliana WUS* genes [16]. According to phylogenetic tree analysis, it can be divided into three clades: the first clade is the modern/WUS clade (*WUS*, *AtWOX1-AtWOX7*), which exists in higher plants; the second clade is the intermediate clade (*AtWOX8*, *AtWOX9*, *AtWOX11* and *AtWOX12*), which originates from tracheophytes; and the third clade is the ancient clade (*AtWOX10*, *AtWOX13* and *AtWOX14*), which originated from phycophyta [17,18]. The ancient origin of the WOX transcription factor and other evolutionary branches derived from plant evolution suggest that this gene family is essential for plant survival.

In the present study, we comprehensively identified the *WOX* genes from the *A. trifoliata* genome. We first determined the *AktWOX* gene structures, motif compositions and chromosomal distributions. Furthermore, we analyzed the phylogenetic relationships and evolutionary patterns in the *AktWOXs*. In addition, the expression patterns of *AktWOXs* during adventitious roots formation were determined. Our results provide insights for further understanding *WOX* family genes in *A. trifoliata*, clarify their evolutionary history, and facilitate their application in gene transformation for improving plants.

## 2. Materials and Methods

### 2.1. Identification and Physicochemical Characterization of AktWOX Sequences

To identify *WOX* genes in the *A. trifoliata* genome, we searched for the conserved HB domain of the corresponding proteins. Public databases including the NCBI Conserved Domain Database (https://www.ncbi.nlm.nih.gov/cdd, 9 August 2022), the SMART database (http://smart.embl-heidelberg.de/, 9 August 2022) and the Pfam database (http://pfam.xfam.org/, 15 August 2022), were used to search the HB domain of candidate sequences, and the domain IDs are PF00046, SM000389 and PF00046 in each database, respectively. Sequences not containing the complete conserved HB domain were removed [19]. After obtaining the *A. trifoliata WOX* genes, the AktWOX protein sequence was submitted to the

conserved domain database (https://www.ncbi.nlm.nih.gov/Structure/bwrpsb/BWRPSB, 25 August 2022) for structural domain filtering to determine the final implant AktWOX transcription factor family members [20]. Gene positions on chromosomes were identified and collinearity mapping was performed using TBtools software [19]. The ExPASy's Prot-Param online tool (http://www.ExPASy.org/tools/protparam.html/, 25 August 2022) was used to predict the physical and chemical properties of the AktWOX transcription factors [21]. We used SOPMA (https://npsa-prabi.ibcp.fr/cgi-bin/, 1 September 2022) [22] to predict the secondary structure of the *WOX* gene in *A. trifoliata*. DataProt-Comp9.0 (http://linux1.softberry.com/berry.phtml?topic=protcomppl&group=Programs&subgroup=proloc, 9 September 2022) and SignalIP5.0 (https://services.healthtech.dtu.dk/service/SignalP-5.0, 15 September 2022) were used for subcellular localization and signal peptide prediction [23].

### 2.2. Sequence Characteristic Analysis, Phylogenetic Analyses, GO Enrichment Analysis and Collinearity of AktWOXs

The evolutionary history was inferred using the Neighbor-Joining method [24]. The optimal tree is shown. The percentage of replicate trees in which the associated taxa clustered together in the bootstrap test (1000 replicates) are shown next to the branches [25]. The evolutionary distances were computed using the Poisson correction method [26] and are expressed in units of the number of amino acid substitutions per site. The proportion of sites where at least 1 unambiguous base is present in at least 1 sequence for each descendent clade is shown next to each internal node in the tree. This analysis involved 10 amino acid sequences. All ambiguous positions were removed for each sequence pair (pairwise deletion option). There were total of 408 positions in the final dataset. Evolutionary analyses were conducted in MEGA 11 software (v11.0.10) [27]. The GFF3 file of the *A. trifoliata* genomic annotation was used to analyze the gene sequence characteristics. GSDS 2.0 (http://gsds.gao-lab.org/, 20 May 2023) was used to count the number and location of exons/introns of the *AktWOXs* [28]. The conserved motifs of the *A. trifoliata* proteins were analyzed by MEME Suite (https://meme-suite.org/meme/tools/meme, accessed on 20 May 2023) [29], where the maximum motif number was set to 10 and the other settings were set to their default values. The above results were subsequently visualized using TBtools [19] software (version 1.0876). To display the evolutionary selection pressure between gene pairs [30], the Ka/Ks ratio was calculated using TBtools [19] software (version 1.0876). The reference genome sequences of *A. thaliana*, *Liriodendron tulipifera*, *Populus x canescens*, *Solanum lycopersicum*, *Glycine max*, *Solanum lycopersicum* and *Amborella trichopoda*, the monocotyledonous plants, *O. sativa*, *Zea mays* and *Andropogon gerardi*, and the Chlorophyta plant *Chlamydomonas reinhardtii* (https://www.ncbi.nlm.nih.gov/, 9 August 2023). We downloaded data from the NCBI database and used them to perform a collinearity analysis with the sequence of *A. trifoliata* [31]. The PlantCARE online website (https://bioinformatics.psb.ugent.be/webtools/plantcare/html/, 26 May 2022) was then used to analyze the cis-acting elements in the 2000 bp promoter region upstream of *A. trifoliata* [32]. Timetree5 (http://timetree.org/, 1 May 2023) was used to reconstruct the evolution of twelve species over time [33]. The Metascape (Metascape.org, 10 May 2023) web-based portal was used for comprehensive gene annotation and analysis resources [33]. A bubble chart was plotted using the Bioinformatics (www.bioinformatics.com.cn, 15 May 2023) free online platform for bioinformatics-related data analysis [34].

### 2.3. Detection AktWOX Existence at Expression Level Using the Public Transcriptomic Data of A. trifoliata Fruit

To further confirm the real existence of *AktWOXs* at the expression level, the transcriptomic data for *A. trifoliata* were downloaded from the NCBI database under BioProject ID PRJNA671772 (https://www.ncbi.nlm.nih.gov/bioproject/PRJNA671772; 25 April 2023) and employed to detect the corresponding expressed sequence. The *A. trifoliata* transcriptomic data contained data on three tissue types (fruit flesh, seeds and rind) at four different stages (young, enlargement, coloring and mature stages), and there were also data for three

biological replicates (young stage, SAMN16551934-36, enlargement stage; SAMN16551937-39, coloring stage; SAMN16551940-42, mature stage). FPKM values calculated by Hisat2 software (v2.1.0) and DESeq2 (v1.36.0) were used to estimate gene expression levels [35].

*2.4. AktWOX Expression during Adventitious Root Formation*

The cuttings used for the experimental treatment were obtained from the same tree cuttings and were exposed to the same cultivation conditions. The cuttings were transplanted in the germplasm nursery of the Sichuan Agricultural University Chongzhou Research Station (30°430 N, 103°650 E); the RNA of 2 cm stem base and root mixed samples at 7, 14, 21, 28, 35, 42, 49 and 56 d during the cutting period of *AktWOX* Shusen 1 was extracted. Total RNA was extracted with an M5 Plant RNeasy Complex Mini Kit (Polysaccharides and Polyphenolics-rich) (JUHEMAI, Beijing, China). The integrity and purity of the RNA were assessed with an Agilent 2100 Bioanalyzer (Agilent Technologies, Santa Clara, CA, USA) and a NanoDrop ND-1000 spectrophotometer (Thermo Scientific, Austin, TX, USA), respectively. Then, the RNA of the samples was reverse transcribed into cDNA using an EasyScript One-Step gDNA Removal and cDNA Synthesis Supermix Kit (TransGen Biotech, Beijing, China).

The primer pairs for the *AktWOXs* and *GAPDH* genes were designed using Primer 3.0 (Table S1), and the primer sequences and related details are listed in Table S3. The amount of cDNA used as the amplification substrate was 1 µmol, and the reaction was carried out as follows: 92 °C for 30 s, followed by 45 cycles of 5 s at 92 °C and 30 s at 53 °C. To determine the expression patterns of the *AktWOXs*, RT-qPCR was conducted on a Thermal Cycler CFX96 Real-Time System (Bio-Rad Laboratories, Hercules, CA, USA) together with PerfectStart Green qPCR SuperMix (TransGen Biotech, Beijing, China). Each sample included three technical replicates. The $2^{-\Delta\Delta Ct}$ method was used to calculate the expression level of genes. Statistical analysis was performed with SPSS (version 20.0.0) and Origin 2018 software (version 9.5.1).

## 3. Results

*3.1. Systemic Characterization of the WOX Gene Family in A. trifoliata*

A total of 10 *WOX* genes were identified from the *A. trifoliata* genome through HMM analysis. They were sequentially named *AktWOX1–9* and *AktWUS* (chromosome 2) (Table 1) according to their positions on the chromosome [36]. The 10 *AktWOXs* had a wide range in gene length (from 657 bp to 11,328 bp) and exon number (from two to five). In terms of protein properties, the 10 *AktWOXs* had obvious differences in amino acid length (from 176 to 361), molecular weight (from 20.500 to 40.173) and isoelectric point (from 5.65 to 9.03). Subcellular localization analysis showed that these proteins were spatially located in the nucleus but had no obvious signal peptide signature.

The secondary structure of 10 AktWOX proteins was predicted and analyzed (Table S2). The α-helical structure and β-folded structure are ordered structures of proteins that have high stability, and random curling is a disordered structure of proteins. The results showed that the 10 *AktWOX* proteins were mainly randomly curled, accounting for 56.28% to 71.75% of the secondary structure, followed by α helices. This indicated that the protein secondary structure of the *AktWOX* family genes was unstable on a whole. The instability coefficient for the proteins in this family was greater than 40, and the hydrophilicity value was less than 0, indicating that they were poorly stable and hydrophilic proteins.

In addition, the corresponding expressed sequences of all 10 *AktWOXs* were detected in the public transcriptomic data of the flesh, seed and rind tissues of *A. trifoliata* fruit (Table S3), which well agreed with the reliability of the *AktWOXs* identified from *A. trifoliata* genome.

**Table 1.** Characteristics of the identified *WOX* gene family members from the *A. trifoliata* genome.

| *WOX* Genes | Gene Length | Chromosome Location | | | Exon | Cell Location | Putative Protein | | | | | |
|---|---|---|---|---|---|---|---|---|---|---|---|---|
| | | | | | | | Length AA | MW (kDa) | PI | Instability Index | Hydrophilic | Signal Peptide |
| *AktWOX1* | 2840 | chr1 | 25,213,082 | 25,215,922 | 5 | Nucleus | 361 | 40.173 | 6.8 | 57.89 | −0.593 | 0.0009 |
| *AktWOX2* | 5392 | chr2 | 896,118 | 901,510 | 3 | Nucleus | 279 | 31.869 | 6.02 | 59.79 | −0.946 | 0.0005 |
| *AktWOX3* | 2204 | chr2 | 7,184,275 | 7,186,479 | 3 | Nucleus | 212 | 24.462 | 9.03 | 52.78 | −1.06 | 0.0023 |
| *AktWOX4* | 11,328 | chr3 | 50,099,349 | 50,110,677 | 3 | Nucleus | 307 | 35.238. | 5.65 | 65.16 | −0.927 | 0.0003 |
| *AktWOX5* | 796 | chr6 | 6,820,221 | 6,821,017 | 2 | Nucleus | 232 | 26.506 | 8.42 | 51.72 | −0.919 | 0.0006 |
| *AktWOX6* | 1120 | chr8 | 2,341,031 | 2,342,151 | 2 | Nucleus | 198 | 23.316 | 8.75 | 61.48 | −0.891 | 0.0003 |
| *AktWOX7* | 1950 | chr9 | 5,460,894 | 5,462,844 | 4 | Nucleus | 316 | 36.673 | 8.44 | 75.25 | −0.949 | 0.015 |
| *AktWOX8* | 2212 | chr15 | 429,281 | 431,493 | 3 | Nucleus | 265 | 29.411 | 5.85 | 60.72 | −0.317 | 0.0007 |
| *AktWOX9* | 657 | chr15 | 27,705,270 | 27,705,927 | 2 | Nucleus | 176 | 20.500 | 8.85 | 53.15 | −0.903 | 0.0044 |
| *AktWUS* | 1622 | chr2 | 2,976,634 | 2,978,256 | 3 | Nucleus | 267 | 29.505 | 6.83 | 60.26 | −0.788 | 0.0012 |

AA, amino acids; PI, isoelectric point; MW, molecular weight. "Instability index" > 40 means unstable; "hydrophilicity" < 0 is hydrophilic, and >0 is hydrophobic.

### 3.2. Phylogenetic Analysis of AktWOX

A phylogenetic tree for the WOX protein family was constructed based on the amino acid sequences of 39 WOX proteins from *A. trifoliata* (10), *O. sativa* (14) and *A. thaliana* (15). According to the evolutionary tree, the 39 WOX proteins were divided into three main branches, and the 10 AktWOXs of *A. trifoliata* were unevenly distributed on the three branches (Figure 1). Among them, six WOX members were assigned to the modern clade, including *AktWUS*, *AktWOX9*, *AktWOX6*, *AktWOX3*, *AktWOX7* and *AktWOX5*. *AktWOX1* and *AktWOX8* were assigned to intermediate clades, and *AktWOX2* and *AktWOX4* were assigned to ancient clades.

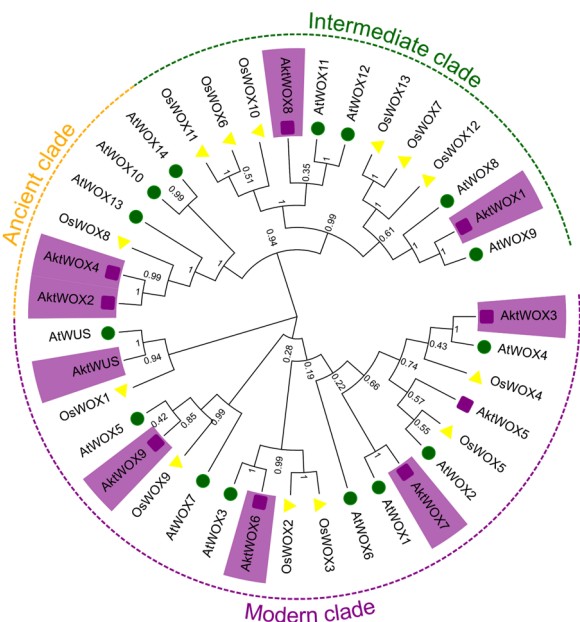

**Figure 1.** Phylogenetic tree analysis of *WOX* genes in *A. trifolium* and other species. *At*: *A. thaliana* (green); *Os*: *O. sativa* (yellow); *Akt*: *A. trifoliata* (purple).

### 3.3. Gene Structure and Conserved Motifs of AktWOXs

Domain analysis showed that the 10 AktWOX protein sequences had conserved HD (Figure S1) and WUS-box domains (Figure S2). Further motif analysis showed that the 10 AktWOX proteins contained 10 relatively conserved motifs (Table S4). Motif 1 and motif 2 were found in all 10 AktWOXs and contained a highly conserved "helix-ring-helix-corner-helix" HD domain (Figure S1). Motif 5 was the WUS-box motif (Figure S2) and existed in *AktWUS*, *AktWOX9*, *AktWOX6*, *AktWOX3*, *AktWOX7* and *AktWOX5* (modern evolution

branch). Motif 7 existed only in *AktWOX1* and *AktWOX8* (intermediate clades), while motif 4 and motif 8 existed only in *AktWOX2* and *AktWOX4* (ancient clades).

An analysis of its exon and intron structure revealed that the *AktWOX* gene contained 2–4 CDSs (coding DNA sequences); *AktWOX3*, *AktWOX5*, *AktWOX6* and *AktWOX9* contained two CDSs; *AktWOX7* had four CDSs; and *AktWUS*, *AktWOX3* and four other members of the middle branch and ancient branch contained three CDSs (Figure 2c).

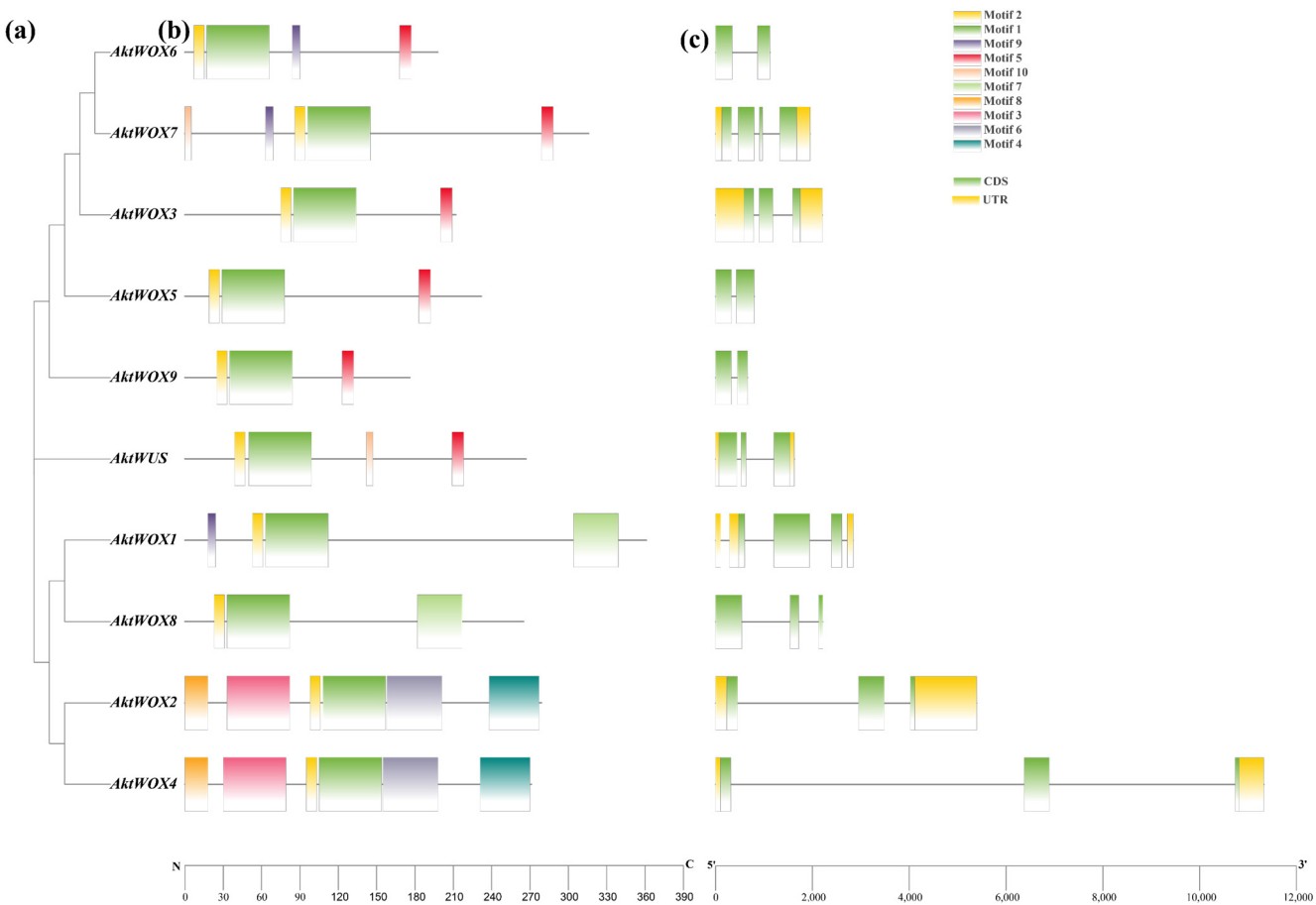

**Figure 2.** Gene and protein structure analyses of the AktWOX family. (**a**) Phylogenetic tree of AktWOXs. (**b**) Motifs of AktWOX proteins. (**c**) Exon-intron structures of *AktWOXs*.

*3.4. Chromosomal Location and Evolutionary Analyses of AktWOXs*

Chromosomal location analysis showed that the 10 *AktWOXs* were distributed on seven chromosomes in *A. trifoliata* (Figure 3), and two *AktWOXs* were located on chromosome 15. The remaining five *AktWOXs* are found on chromosomes 1, 3, 6, 8 and 9.

In terms of evolution, intraspecies collinearity analysis showed that dispersed and segmental or whole-genome duplication (WGD) events were the main sources of *AktWOX* expansion (Figure 3), but the majority (eight; 80%) *AktWOXs* were derived from dispersed replication, and the minority (2; 20%) *AktWOXs* were derived from WGD events.

To further understand the gene duplication mechanism in the *WOX* gene family in *A. trifoliata*, a comparative map was generated using the dicotyledonous plants *A. thaliana*, *L. tulipifera*, *Populus* × *canescens*, *S. lycopersicum*, *G. max*, *S. tuberosum* and *A. trichopoda* and the monocotyledonous plants *O. sativa*, *Z. mays* and *A. gerardi*. They were analyzed with the Chlorophyta plant *C. reinhardtii* (Figure 4). The number of homologs between *A. trifoliata* and *A. thaliana* was eight, the number in *L. tulipifera* was twleve, in *Populus x canescens* it was eighteen, in *S. lycopersicum* it was seven, in *G. max* it was nineteen, in *S. tuberosum* and *A. trichopoda* there were six and six, in the monocotyledonous plant *O. sativa* there was three, in *Z. mays* the number was five, in *A. gerardi* it was fourteen and the

chlorophyta plant *C. reinhardtii* did not contain any homologs, indicating a strong direct homology between the *A. trifoliata WOXs* and the dicotyledons members, which showed a high degree of evolutionary divergence compared with the monocotyledons. Table S5 shows a Synteny analysis of WOX genes between *A. trifoliata* and other plants.

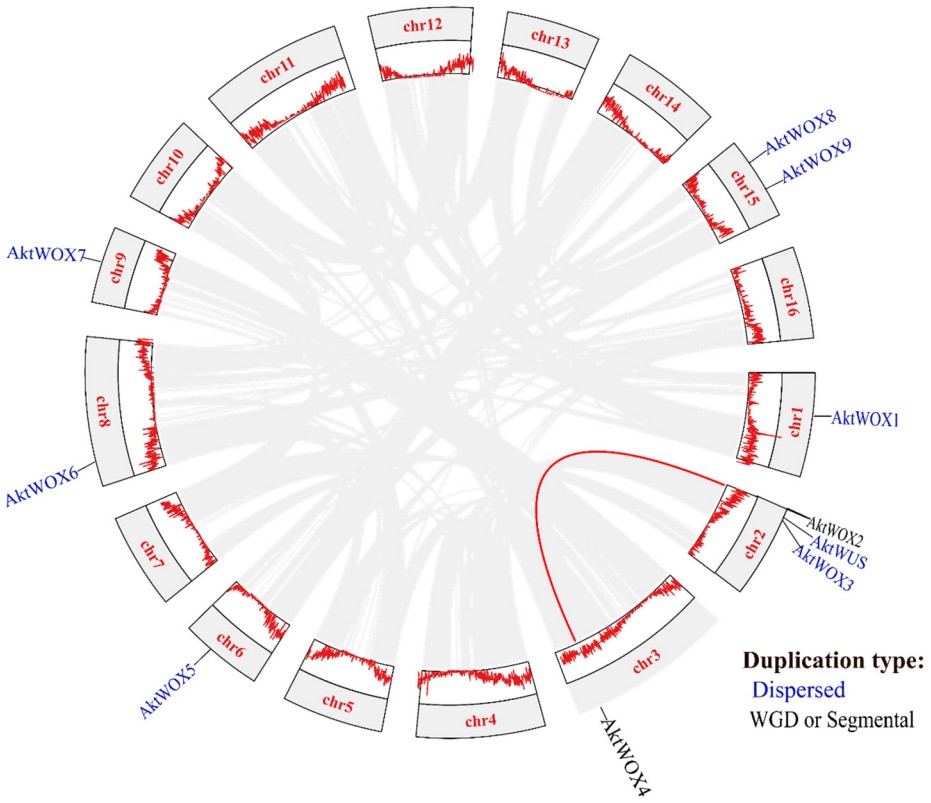

**Figure 3.** Collinearity and gene duplication events and gene clusters in *AktWOXs*. The red line indicates the *AktWOX* collinear gene pair; the two gene duplication types (dispersed, WGD or segmental) are represented in blue and black, respectively.

Determining the Ka/Ks ratio can effectively improve the understanding of the evolutionary constraints of the *WOX* gene family. The Ka/Ks values of all 45 homologous *AktWOX* pairs were much lower than 1 and varied from 0.01 to 0.33 (Table S6), indicating that the *AktWOXs* could have experienced a strong purifying selection during their evolutionary history.

*3.5. Identification of Cis-Acting Elements of the AktWOX Gene Family*

The cis-element analysis results for the upstream sequence of *AktWOXs* are shown in Figure 5. The types of *AktWOX* cis-elements included hormone-responsive elements and environment-responsive elements, and each element had five and seven subtypes, respectively.

There were three cis-acting elements related to stress resistance: defense and stress response elements (13), low-temperature induction response elements (8), light responsiveness elements (131), anaerobic induction elements (22), zein metabolism regulation elements (9), elements involved in endosperm expression (9) and elements involved in flavonoid biosynthesis genes (2). Cis-acting elements related to hormone regulation mainly included auxin (7), gibberellin (11), abscisic acid (33), MeJA responsiveness elements (42) and salicylic acid response elements (8). Cis-acting elements related to substance synthesis: elements involved in metabolic regulation of zein, elements involved in endosperm expression and MYB-binding site elements involved in flavonoid biosynthesis genes.

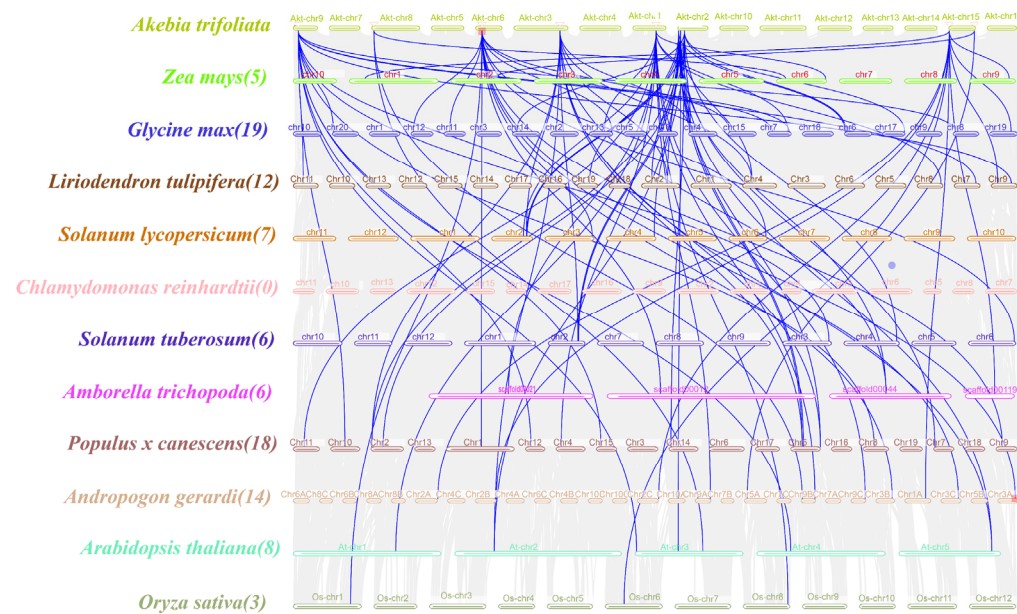

**Figure 4.** Collinearity analysis between *WOX* genes in *A. trifoliata* and *WOX* genes in other species. Different species names and chromosomes are represented by different colors. The blue line indicates the homologous *WOX* gene pairs between other species and *AktWOXs*, and the number in parentheses after the species name indicates the number of collinear pairs between the *WOX* genes of the other species and *AktWOXs*.

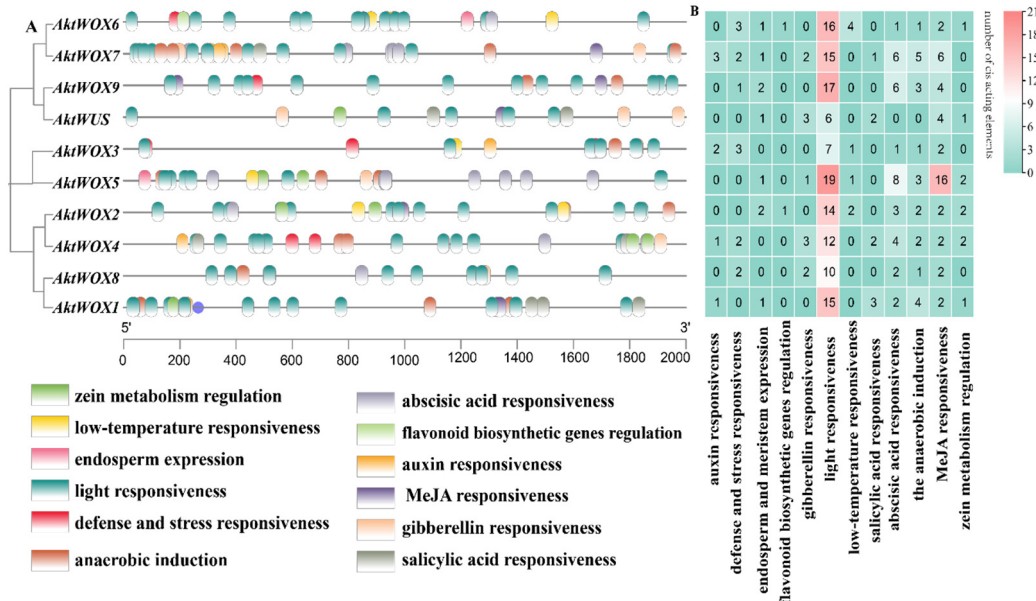

**Figure 5.** Prediction of cis-elements in the *WOX* promoter of *A. trifoliata*. (**A**) The distribution of cis-acting elements in the 2000 bp region upstream of the transcription start site of *AktWOXs*; (**B**) the number of cis-acting elements of the two functional categories in *AktWOXs*, respectively, indicated by different colors and numbers, cyan-white-red represents the increasing number of cis-acting elements.

Both the type and the number of cis-acting elements also widely varied among members of the *AktWOXs* (Table S7). We found that every *AktWOX* had a light-responsive element with numbers ranging from six to nineteen, and *AktWOX5* and *AktWUS* had the most (51) and least (17) cis-acting elements, respectively. The number of cis-acting element subtypes varied from six (*AktWUS*, *AktWOX8* and *AktWOX9*) to nine (*AktWOX4*, *AktWOX6*

and *AktWOX7*), and the *AktWOX3* genes contained seven cis-acting element subtypes, and the *AktWOX1*, *AktWOX2* and *AktWOX5* genes contained eight cis-acting element subtypes.

### 3.6. GO Enrichment Analysis of AktWOX Genes

The 10 *AktWOX* genes were divided into three categories (Figure 6), molecular functions (MFs), cellular components (CCs) and biological processes (BPs), by GO enrichment analysis, with nine, twelve, and one hundred eighty-three subcategories (Table S8), respectively (Figure 6). Eight *AktWOX* genes were involved in MFs, such as transcriptional regulatory activity and DNA-binding transcription factor activity; four were involved in CCs; and seven were involved in BPs, such as RNA biosynthesis, the regulation of cell metabolism, the regulation of biosynthesis, nucleic acid metabolism and transcription regulation.

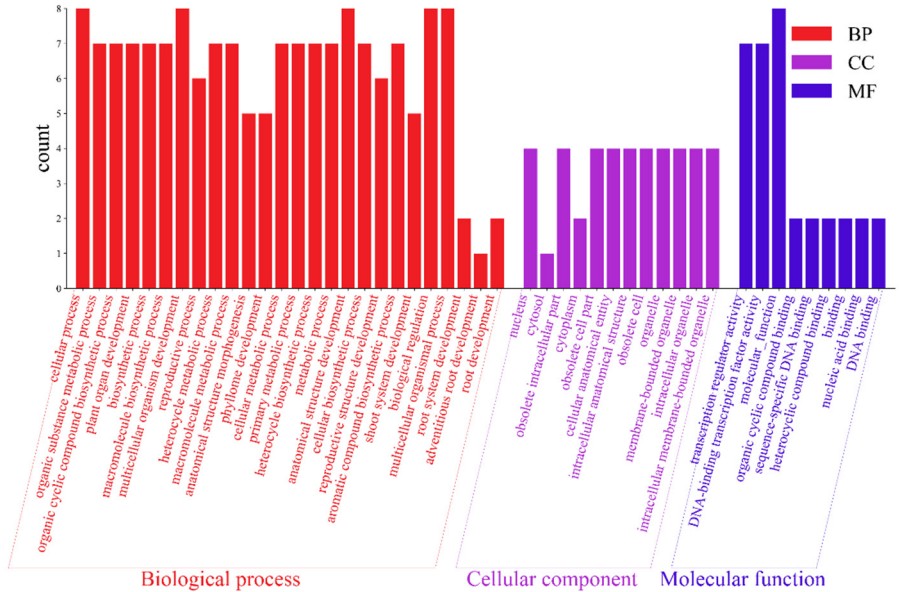

**Figure 6.** GO function analysis histogram.

### 3.7. AktWOXs Expression during the Growth of Adventitious Roots of A. trifoliata

Sequence homology alignment revealed that six *AktWOX* genes (*AktWOX1*, *AktWOX2*, *AktWOX3*, *AktWOX4*, *AktWOX8* and *AktWOX9*) were homologous to *AtWOX4*, *AtWOX5*, *AtWOX7*, *AtWOX9*, *AtWOX11*, *AtWOX12*, *AtWOX13* and *AtWOX14* in *A. thaliana* [37]. We further examined the expression of these six genes during the growth of adventitious roots of *A. trifoliata*. These genes are reportedly related to root growth and development.

The RT-qPCR results showed that the expression of the *WOX* genes during adventitious root formation of *AktWOX1*, *AktWOX2*, *AktWOX8* and *AktWOX9* increased to the highest values at 42 d of development (Figure 7), and then their expression gradually decreased to below the initial levels. Their expression increased at 28 d and decreased slightly with increasing development time but was still above the initial level. *AktWOX2*, *AktWOX3* and *AktWOX4* expression patterns were similar throughout adventitious root formation (Figure 7), with their expression decreasing at the beginning of development, being rapidly upregulated at 35 d and continuing until 42 d, after which their expression again decreased. Figure S3 amplification and dissolution curves of qRT-PCR, all data for statistical analysis are presented in Supplementary Materials Table S9.

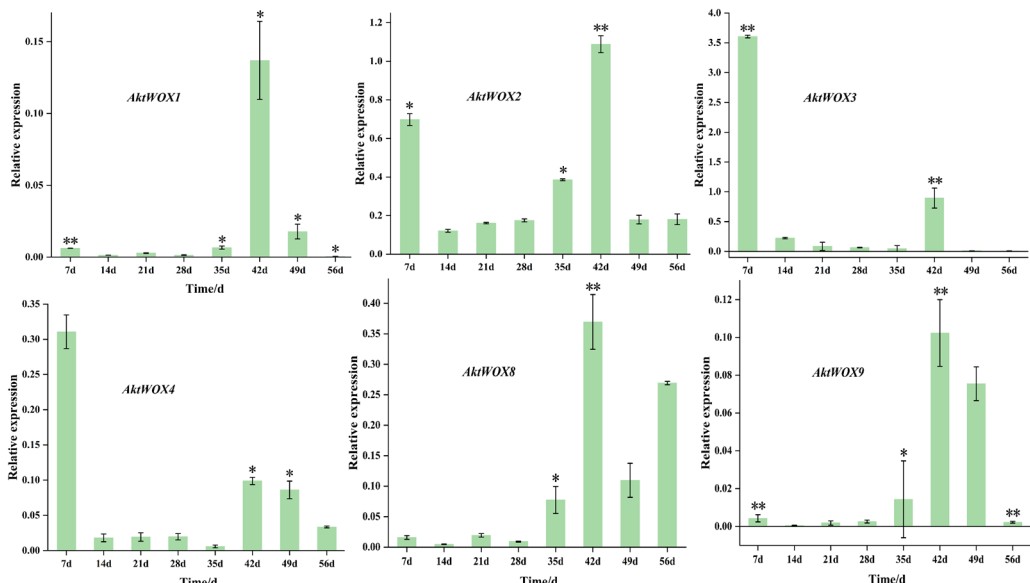

**Figure 7.** qRT−PCR analysis of the expression of 6 *AktWOXs*. Transcript levels of *AktWOXs* were calculated using the $2^{-\Delta\Delta Ct}$ method. The error bars represent the standard deviations of three replicates. * represents the significance between peak expression the expression at 0 d (**, $p < 0.01$).

## 4. Discussion

### 4.1. The AktWOX Gene Structure Is Extremely Conserved during Evolution

In plants, the *WOX* family is an extremely important gene family, and the proteins it encodes are involved in the growth and development of almost every organ within angiosperms [38]. As an increasing number of plant genomes are sequenced and released, many *WOX* genes in angiosperms have been systematically identified and studied. Enriching the number of reported *WOX* genes in basal dicots will further improve our understanding of the evolution of the *WOX* gene family [37]. In this study, we identified 10 *WOX* genes from the genome of the basal dicot *A. trifoliata*. We found that, although the *AktWOX* genes have wide differences at the DNA sequence level, mainly reflected in the number of introns and sequence length, the physical and chemical properties of the proteins they encode are extremely similar, including similar protein lengths and molecular weights and the instability and hydrophilicity of the protein structure (Table 1). Moreover, these characteristics of *WOX* genes in *A. trifoliata* are very similar to those of monocots, including wheat [15] and four Euphorbiaceae plants [39], as well as core dicots, including *A. thaliana* [16] and *Phaseolus vulgaris* [40]. This indicates that *WOX* genes are highly conserved during evolution, and the gain and loss of introns/exons are the driving forces for the evolution of this gene family.

From an evolutionary perspective, genes duplicated by different mechanisms, such as WGDs, and tandem and dispersed duplications, are primary raw materials for new gene origins and evolution and ultimately result in functional novelty and specialization [41]. Some studies have shown that, following WGD events, genes encoding TFs are preferentially retained [42]. Two WGD events occurred in *A. trifoliata* approximately 85 and 140 million years ago (θ event), respectively. The former is a specific WGD event in *A. trifoliata*, and the latter occurred during the early stages of dicotyledonous plant differentiation (θ event) [43]. In this study, eight (80%) of ten identified *AktWOXs* were found to be derived from dispersed duplication, two *AktWOXs* were found to be derived from WGD (Figure 3), which suggested that dispersal was the major force of *AktWOX* origin, and the *AktWOX* gene family was involved in only one WGD event. We reconstructed the evolution of twelve species over time (Figure S4) and showed that the *AktWOX* gene family was involved in a specific genome-wide duplication event in *A. trifoliata*. In addition, the fact that all Ka/Ks values of the homologous *AktWOX* pairs were much lower than

one (Table S5) further suggested that all *AktWOXs* experienced strong purifying selection during their evolutionary history. The Ka/Ks value of two combinations between *AktWUS* and both *AkWOX2* and *AktWOX4* was very close to 0.004, while the combination (*AktWUS* and *AktWOX8*) with the largest Ka/Ks value was also related to *AktWUS* (Table S5), which indicated that *AktWUS* could be an ancestral gene of the *AktWOX* family. This evolutionary evidence further demonstrates that *AktWOXs* are highly conserved.

*4.2. AktWOX Gene Family Members May Have Greatly Diverged Functions*

Many reports have confirmed that WOX transcription factors play important roles in regulating plant growth and development, including embryonic development, maintenance of meristematic stem cells, seed formation, regeneration of isolated tissues and organs and response to abiotic stress. For instance, *WOX* genes play different roles in the development of *O. sativa* roots, stems and leaves [44]. The *OsWOX6* gene plays a major role in the regulation of seed development, especially for the growth and development of seeds under water-deficient conditions [45]. *WOX* genes are widely involved in the growth and development of different plant organs as well as physiological and biochemical processes, but their protein structure and gene number are very conserved, which indicates that this gene family has extensive functional differentiation. The function and expression pattern between the members of a gene family in plants have changed based on their upstream regulatory regions, such as the promoter, or mutation in the coding region during evolution, and these changes cause them to participate in different processes and pathways [46,47]. In this study, sequence analysis of the *AktWOX* promoter results showed that *AktWOXs* not only play important roles in the response to light signals and resistance to stress but also play a role in endosperm or seed development and meristem formation. GO enrichment analysis indicated seven *AktWOXs* widely involved in various growth development and tissue metabolism processes in biological processes of *A. trifoliata*. The results indicate that the *AktWOX* gene family is functionally diverse.

The HD and WUS-box domains are two conserved domains in the WOX family [16]. At present, research on the WUS-box domain is mainly based on the *WUS* gene. In the process of WUS participating in maintaining the characteristics of stem cells in the plant stem meristem, the WUS box mainly exerts inhibitory activity and maintains the dynamic balance of stem cell proliferation regulation [48]. Studies have shown that the WUS box plays an important role in the maintenance of stem cell characteristics [48]. Therefore, the modern branch of *AktWOXs* may be involved in regulating the development of stem cells. This is further evidence that members of the *AktWOX* gene family may have wide functional differences.

*4.3. The AktWOX Gene May Be Involved in Adventitious Root Regulation*

The *WOX* gene family is widely involved in the formation of adventitious roots. In *A. thaliana*, *AtWOX4*, *AtWOX5*, *AtWOX7*, *AtWOX9*, *AtWOX11*, *AtWOX12*, *AtWOX13* and *AtWOX14* are associated with root growth and development [36], while there are no homologous genes for *AtWOX7*, *AtWOX12* and *AtWOX14* in *AktWOX*. De novo root organogenesis from tissue explants requires consecutive cell fate transition steps to finally form an adventitious root. The first step of cell fate transition is priming, which results in the formation of adventitious root founder cells. The second step of cell fate transition is initiation, which results in the formation of the dome-shaped root primordium via cell division. The expression levels of *AtWOX11/12* decrease and those of *AtWOX5/7* increase as the root founder cells transition into the root primordium [9]. In the formation of adventive roots of *A. trifoliata*, *AktWOX2* and *AktWOX8* were highly expressed in the late stage, and *AktWOX3* and *AktWOX4* were highly expressed in the early stage. This result, which is similar to that of Hu et al. [9], may indicate that *AktWOX3* and *AktWOX4* are related to the initiation of adventitia root cells in *A. trifoliata*, while *AktWOX2* and *AktWOX8* may be related to the initiation of adventitious root cells.

## 5. Conclusions

In this study, we identified 10 candidate *AktWOXs* that were unevenly distributed on seven high-quality assembled chromosomes in the *A. trifoliata* genome. All 10 *AktWOXs* were classified into three groups, and in terms of evolution, they were mainly produced by dispersal events and underwent strong purifying selection. We further identified four genes, namely, *AktWOX2*, *AktWOX3*, *AktWOX4* and *AktWOX8*, that could be involved in the response to adventitious root formation conditions. In addition, this study provides important information concerning the *WOX* genes of *A. trifoliata* and provides a theoretical reference for their functions in adventitious root formation.

**Supplementary Materials:** The following supporting information can be downloaded at: https://www.mdpi.com/article/10.3390/cimb46010002/s1. Figure S1: Alignment of the homeodomain sequences. Figure S2: Alignment of the WUS box that is located downstream of the homeodomain. Figure S3: Amplification and dissolution curves of qRT-PCR. Figure S4: The evolution of twelve species over time. Table S1: Specific primer sequence for WOX genes of *A. trifoliata*. Table S2: secondary structure prediction. Table S3: The transcriptomic data of *A. trifoliata*. Table S4: Conserved motifs of WOX protein in *A. trifoliata*. Table S5: Synteny analysis of WOX genes between *A. trifoliata* and other plants. Table S6: The value of Ka/Ks. Table S7: Putative cis-elements in the promoter of AktWOXs. Table S8: GO enrich Result. Table S9: qRT−PCR analysis of the expression of some AktWOXs.

**Author Contributions:** S.C. and P.L. wrote the manuscript; S.C. and P.L. conceived and designed the research project; Y.Z. and H.Y. collected the materials; C.C. carried out the experiments and analyzed the data; T.R. and F.T. provided the experimental reagents and analytical tools. All the authors contributed to the revisions and comments on the manuscript. All authors have read and agreed to the published version of the manuscript.

**Funding:** This research was supported by the Science and Technology Department of Sichuan Province, grant numbers 2022ZHXC0002, 2022ZHXC0028 and 2022ZHXC0044, City school cooperation project of Ya'an City Science and Technology Bureau: 22SXHZ0071, Sichuan Administration of Traditional Chinese Medicine: 2023zd026.

**Institutional Review Board Statement:** Not applicable.

**Informed Consent Statement:** Not applicable.

**Data Availability Statement:** All data analyzed during this study are included in the manuscript and Supplementary Materials. Genome sequence files of *A. trifoliata* were downloaded from the National Genomics Data Center database under BioProject PRJCA003847. Transcriptome data for *A. trifoliata* were downloaded from the NCBI database under accession numbers PRJNA671772, SAMN16551931–33, SAMN16551934–36, SAMN16551937–39 SAMN16551940–42 and the National Genomics Data Center database under BioProject PRJCA014987.

**Acknowledgments:** We are very grateful to the editor and reviewers for critically evaluating the manuscript and providing constructive comments for its improvement.

**Conflicts of Interest:** The authors declare no conflict of interest.

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
