# Peer review of "Global Analysis of the WOX Transcription Factor Family in Akebia trifoliata"

_cimb, doi:10.3390/cimb46010002_

Round 1

Reviewer 1 Report

Comments and Suggestions for Authors

Manuscript "Global Analysis of the WOX Transcription Factor Family in Akebia trifoliata" is very interesting.

General comments:
Authors identified the WOX genes from the A. trifoliata genome. Authors determined the AktWOX gene structures, motif compositions, and chromosomal distributions and analyzed the phylogenetic relationships and evolutionary patterns of the AktWOXs. Authors determined expression patterns of AktWOXs in different A. trifoliata tissues and under different AR formation conditions.

Detailed comments:
Introduction section is perfect.
The authors used a hidden Markov model. However, they did not provide any input parameters or the form of the model according to which the Bayesian analyses were performed.
A method of dendrogram construction is given. However, information on the method of counting similarity/difference was missing.
The description of the methodology lists a number of programs used. However, this is not a description of the methodology. First of all, the methods used to achieve the objectives of the work should be given.
Figure 5 should be supplemented with the results of statistical analysis.
Figure 6 should be supplemented with the results of statistical analysis.
Figure 7 should be supplemented with the results of statistical analysis.
Figure 8: regression curves are given. Models and determination coefficients should be supplemented. What characteristics are indicated by the bars used.

My suggestions:
"A. trifoliata" - italic
"2−ΔΔCt" - "-ΔΔCt" should be in superscript.

Paper needs major revision.

Reviewer 2 Report

Comments and Suggestions for Authors

Manuscript entitled “Global analysis …… Akebia trifoliata” described characterization of WOX transcription factor. Authors identified a total of 10 WOX-TF using in-silico tools. They claimed that WOX-TF expressed during fruit development and adventitious root formation, and thus involved in the development of the three fruit tissues, flesh, seeds and rind, and play an important role in the regulation of adventitious root development.

My observations/suggestions are:–

1.      Section 2.3: Authors used in-silico means for the expression analysis of WOX-TF in fruit development. It is general practice that such type of observation is always supported by the real-time based experiments. My suggestion is to verify these observations, a wet-lab experiment is needed. Authors have to design primers and perform real-time expression study during fruit development to verify the in-silico data. (Similarly, as done in section 2.4).

2.      How authors analyzed presence of intron-exon. Generally, TF are short and intron-less.

3.      Real-time data should be backed/supported by melt-curve analysis to confirm the primer specificity.

Round 2

Reviewer 1 Report

Comments and Suggestions for Authors

The authors made changes based on my suggestions. I recommend publishing the manuscript in its current form.

Author Response

Thank you again for your comment.

Reviewer 2 Report

Comments and Suggestions for Authors

I do not agree with the response of comment-01. I do not understand why you need April and August. Simple you have to design primers, and do real-time PCR (after treatment; plants can grow in culture room/growth chambers).

Second, why you stated that "experiment cannot be repeated"  and "experimental design cannot be realized".

Author Response

We have made the modifications. For detailed information please see the attachment.

Round 3

Reviewer 2 Report

Comments and Suggestions for Authors

Authors response is OK